# The Effectiveness of Individual or Group Physiotherapy in the Management of Sub-Acromial Impingement: A Randomised Controlled Trial and Health Economic Analysis

**DOI:** 10.3390/ijerph17155565

**Published:** 2020-08-01

**Authors:** Ian Ryans, Rhona Galway, Annette Harte, Rejina Verghis, Ashley Agus, Neil Heron, Roland McKane

**Affiliations:** 1General Practice Elective Care Service (MSK), Eastern GP Federations, Belfast, BT8 7AR, Ireland; 2Rheumatology Department, South Eastern Health and Social Services Trust, Dundonald BT16 1RH, Ireland; rhona.galway@setrust.hscni.net (R.G.); Roland.McKane@setrust.hscni.net (R.M.); 3School of Health Sciences, Ulster University, Belfast BT37 0QB, Ireland; aa.harte@ulster.ac.uk; 4Northern Ireland Clinical Trials Unit, Belfast BT12 6BA, Ireland; rverghis01@qub.ac.uk (R.V.); ashley.agus@nictu.hscni.net (A.A.); 5Department of General Practice, Queen’s University Belfast, Belfast BT7 1NN, UK; N.Heron@qub.ac.uk; 6Department of Primary Care, Keele University, Staffordshire ST5 5BG, UK

**Keywords:** shoulder, physiotherapy, group, subacromial impingement

## Abstract

*Background*: Shoulder pain is common in primary care. The management of subacromial impingement (SAI) can include corticosteroid injections and physiotherapy. Physiotherapy can be on an individual or group basis. *Aim*: To examine the clinical effectiveness and make an economic analysis of individual versus group physiotherapy, following corticosteroid injection for SAI. *Design and Setting*: A single-blind, open-label, randomised equivalence study comparing group and individual physiotherapy. Patients referred by local general practitioners and physiotherapists were considered for inclusion. *Method*: Patients were randomised to individual or group physiotherapy groups, and all received corticosteroid injection before physiotherapy. The primary outcome measure was shoulder pain and disability index (SPADI) at 26 weeks. An economic analysis was conducted. *Results and Conclusion*: 136 patients were recruited, 68 randomised to each group. Recruitment was 68% of the target 200 participants. SPADI (from baseline to 26 weeks) demonstrated a difference (SE) in mean change between groups of −0.43 (5.7) (*p*-value = 0.050001), and the TOST (two-one-sided test for equivalence) 90% CI for this difference was (−10.0 to 9.14). This was borderline. In a secondary analysis using inputted data, patients without SPADI at week 26 were analysed by carrying forward scores at week 12 (mean difference (95% CI) = −0.14 (−7.5 to 7.3), *p*-value = 0.014). There is little difference in outcome at 26 weeks. Group physiotherapy was cheaper to deliver per patient (£252 versus £84). Group physiotherapy for SAI produces similar clinical outcomes to individual physiotherapy with potential cost savings. Due to low recruitment to our study, firm conclusions are difficult and further research is required to give a definitive answer to this research question. (NCT Clinical Trial Registration Number NCT04058522).

## 1. Background

Shoulder pain is a common problem in the primary care population [1] causing significant disability and morbidity [2,3,4]. There are numerous clinical diagnostic categories described for shoulder pain. However, subacromial impingement (SAI) is the most common category of shoulder pain, encompassing several pathologies that reduce the subacromial space [5,6,7].

In primary care, SAI is commonly treated by corticosteroid injections and/or physiotherapy. However, evidence for the effectiveness of these strategies is unclear [8,9,10]. Low quality evidence suggests that subacromial corticosteroid injections have a small, positive short-term effect [9]. One review suggests that exercises, as typically prescribed by physiotherapists [6], may be as effective as surgery. Published guidelines state that exercises focused on the rotator cuff and scapular stabilizers are more effective than general exercise; manual joint mobilisations bring no additional benefit when added to an exercise regime, but soft tissue release/massage is more effective than placebo [11]. A study [12] examining the cost-effectiveness of injection plus exercises compared to exercises alone for SAI reported that, despite injection and exercises being more expensive, it may be a cost-effective strategy compared with exercise alone. A review of the effectiveness of manual therapy and exercise suggested that these may be similar in effectiveness to subacromial injection [10]. Further evidence is therefore required to improve the management of SAI examining novel approaches or combinations of the above treatments [10].

Physiotherapy is often given on a one-to-one basis. However, many conditions are managed in group settings [12], reducing costs and providing peer support for patients. Studies evaluating group-based physiotherapy for musculoskeletal conditions have revealed clinically irrelevant differences [12], but there is little evidence examining this approach for shoulder pain.

The aim of this study was to review the equivalence of outcome between group and individual physiotherapy in the treatment of SAI. An economic analysis was also conducted to compare health service use and associated costs of the two groups.

The hypothesis is that physiotherapy treatment for shoulder impingement delivered as a series of six rehabilitation classes will be as effective as individual physiotherapy treatment.

## 2. Methods

### 2.1. Study Design

This was a single-blind, open-label, randomised equivalence study comparing group and individual physiotherapy. Ethical approval was obtained from the Office for Research Ethics Committee, Northern Ireland (07/NIR01/79). The study was registered retrospectively at clinicatrails.gov (Registration number NCT04058522).

### 2.2. Population

Patients referred by local general practitioners (GPs) and physiotherapists were considered for inclusion. The patients included in the trial are therefore typical of primary care patients referred to physiotherapy for management of their SAI symptoms. A physiotherapist (R.G.) screened patients for inclusion and exclusion criteria (Table 1), obtained written informed consent approved by the ethical committee and randomised them to treatment groups.

The Consolidated Standards of Reporting Trials (CONSORT) diagram is presented in Figure 1. Data was not registered for patients assessed for eligibility but not recruited to the trial as this was not considered at the time.

### 2.3. Intervention

Six rotator cuff rehabilitation classes: one class per week for six weeks (30 min length) aiming for 5–10 participants. Classes included advice on the condition, exercises for scapulo-humeral mobility and stability, and specific rotator cuff rehabilitation exercises. One experienced band 6 physiotherapist supervised each of the classes.

### 2.4. Comparator

Routine individual physiotherapy: one session per week for six weeks (30 min). Treatment was based on evidence-based guidelines for the treatment of SAI (CSP 2005) and consisted of mobilisation techniques, supervised exercises and stretches. The physiotherapists taking the individual sessions were of the same level of experience as in the group intervention and were able to select appropriate treatments for each patient from this protocol based on their clinical judgement.

### 2.5. Steroid Injection

After randomisation all patients in both intervention and control groups received a single steroid injection of Triamcinolone 40 mgs (1 mL) through a lateral approach to the subacromial space. This has been shown to give short term pain relief [9], in order to facilitate patients to undertake a rehabilitation programme. All injections were successfully administered, anatomically guided and performed by the same GP (IR) experienced in musculoskeletal medicine, blinded to group allocation. An interval of between one and three weeks was allowed between injection and commencement of physiotherapy.

### 2.6. Outcome Measures

Primary outcome: The primary outcome measure was the SPADI (shoulder pain and disability index) at 26 weeks. The SPADI consists of 13 items that assess two domains: a five-item subscale that measures pain (range 0–50) and an eight-item subscale that measures disability (range 0–80). A higher score indicates greater impairment or disability [14]. The SPADI has previously been used in primary care studies, including in rotator cuff disease [15]. A systematic review of the SPADI has found reliability coefficients of ICC ≥ 0.89 in a variety of patient populations [16].

Secondary outcomes: Secondary outcomes were the active range of external rotation in neutral position (measured by goniometer) and internal rotation (distance (centimetres)) between thumb tip and C7 spinous process), global patient self-assessment measured by a 100mm visual analogue scale, and general health status with Short Form 36-item version 2 (SF36v2) [17], EuroQol 5-dimension 5-level (EQ-5D-3L) [18] and Hospital Anxiety and Depression Scale (HADS) [19].

Outcomes were recorded at baseline, weeks 12, 26 and 52 by one researcher (R.G.) who was blinded to all procedures.

### 2.7. Study Withdrawals

Patients were withdrawn from the study if they received additional steroid injections during the study or if they were referred for ultrasound guided steroid injection at 12 or 26 weeks review appointments. No further data was collected from patients withdrawn from the study.

### 2.8. Randomisation, Allocation Concealment and Blinding

The block randomisation technique was used to generate the randomisation schedule. The schedule was generated using Microsoft Excel and the block size was 20. The randomisation allocations were placed in opaque, sequentially numbered, sealed envelopes to ensure allocation concealment. The envelopes were opened sequentially, after receiving patient consent. The outcome assessors were blinded to the treatment allocations, and patients were instructed not to reveal their treatment group at face to face assessments.

### 2.9. Sample Size Calculation

This was an equivalence trial. In order to detect the minimum clinically significant change of ±10 points [20] on the SPADI scale at 26 weeks, assuming a standard deviation of 23.5, with 80% power and a significance level of 5% for a two-tailed analysis, 100 subjects per group was required, allowing for 5% drop out rate.

### 2.10. Statistical Analysis

The primary outcome measure was the change in SPADI score between baseline and week 26. The equivalence margin was set at (−10, +10) as this is reported as a clinically significant change in Total SPADI score [20]. The 95% CI of mean difference (*µ*_1_*−µ*_2_) in Total SPADI score within the range of −10 to +10 is therefore considered as the proof for the equivalence of both interventions. The null hypothesis was that the treatments are not equivalent, i.e., (*µ*_1_*−µ*_2_) <= −10 or (*µ*_1_*−µ*_2_) >= 10_._ A two one sided test (TOST) was used to test the hypothesis. The number of patients with 26 weeks SPADI was low. A post-hoc analysis was carried out imputing the week 12 values for the patients who did not have 26 week data. This was not included in the initial plan.

Secondary outcomes were summarised using *n*, Mean (SD), and Median (IQR). The difference between the groups were compared using analysis of covariance, adjusting for the baseline values. Stata 12.0 for Windows (Stata Corporation, College Station, TX, USA) and R 3.0.2 (R Foundation for Statistical Computing. Vienna, Austria) was used for the data analysis.

### 2.11. Health Economic Analysis

Patient use of healthcare services related to their shoulder pain was collected at the 12, 26 and 52 week attendances by use of follow-up questionnaires. This included: intervention and control physiotherapy attendance, contacts with general practitioner, accident and emergency visits, additional physiotherapy, hospital investigations, outpatient attendances, analgesia and Non-Steroidal Anti-inflammatory Drugs (NSAIDs) use, and other medications. The quantity of resource used was multiplied by the appropriate unit costs. Steroid injections were not included as this was a cost common to both arms of the trial.

Total health service cost was calculated by adding costs incurred at weeks 12, 26 and 52. Since healthcare costs are typically skewed, non-parametric bootstrapping was also used to calculate 95% bootstrap confidence intervals of differential mean costs.

## 3. Results

### 3.1. Baseline Characteristics

A total of 136 patients in the age range 28.5 to 84.1 years was recruited to the study from September 2008 to February 2012. Recruitment was slow and was closed short of the expected recruitment of 200 due to time constraints on the researchers and funder. Baseline characteristics showed that patients were predominantly female (61%), 83% were right arm dominant and 66% had right shoulder pain. (Table 2) Symptoms had been present for a mean of 30 weeks in the intervention group and 33.5 weeks in the control group. A small proportion reported shoulder pain post injury (26%), with 28% seeking previous injections and 31% receiving previous physiotherapy. Effects of both these interventions were not long lasting and with 45% in paid employment there was a considerable loss of working days due to shoulder pain. There were no significant differences in baseline characteristics between the groups.

### 3.2. Primary Outcome Measure

Table 3 summarises the SPADI Pain, Disability, and Total score at baseline, 12, 26 and 52 weeks. The SPADI score in both the intervention and control arms indicated a reduction in all domains, with the lowest scores at 52 weeks. The overall SPADI (from baseline to 26 weeks) demonstrated a difference (SE) in mean change between the two groups of −0.43 (5.7) (*p*-value = 0.050001). The TOST (two one-sided test for equivalence) 95% CI for this difference is given as (−10.0 to 9.14), which includes −10, supporting the null hypothesis that the groups are not equivalent, i.e., the intervention (those receiving group physiotherapy) have improved more than the control group (individual physiotherapy) (Table 4).

As the number completing SPADI at 26 weeks was low, secondary analysis was carried out. Patients without SPADI data at week 26 were analysed by carrying forward scores obtained at week 12 to calculate the mean difference (SE) in the change between the groups. The null hypothesis that the groups were not equivalent was rejected (mean difference (95% CI) = −0.14 (−7.5 to 7.3), *p*-value = 0.014). Figure 2 shows the mean and 95% CI in SPADI Total score between the treatment arms at week 12, 26, 52. The mean and CI overlap significantly, implying that the SPADI remains similar in both groups.

### 3.3. Secondary Outcome Measures

Both groups showed improvement in secondary outcome measures but there was no significant difference between groups, except for external rotation at 26 weeks favouring the control group (*p* = 0.05).

For General Health Status, SF36v2, EQ-5D-3L and HAD, the only significant difference was at 12 weeks in the EQ-5D-3L favouring the intervention group (group physiotherapy) (*p* = 0.01).

### 3.4. Physiotherapy

On average, both group and individual sessions lasted 30 min. There was no significant difference in the number of sessions attended in both groups (mean sessions attended were 5.5 in the intervention (group physiotherapy) group and 5 in the control (individual physiotherapy) group). Mean number of participants per group was three. One patient in the intervention arm received routine physiotherapy session instead of group session and three patients received group sessions instead of individual physiotherapy due to an administration error in the physiotherapy department.

### 3.5. Health Economic Analysis

Not all patients received the physiotherapy treatment that they were randomised to receive (see Section 3.4), and not all patients attended all six sessions. We therefore only costed for what they actually received. Individual physiotherapy was costed at £42 per session (see Table 5) and since it was observed in the trial that an average of three patients attended the group physiotherapy (intervention) sessions, we costed these at £14 per group session. Table 6 presents health service use costs at each timepoint and Table 7 presents physiotherapy treatment costs and total costs (health service costs and physiotherapy costs) related to shoulder pain. Medication costs were not included in the analysis as this was poorly recorded. Health service use was slightly higher in the intervention arm, but differences were small, and 95% confidence intervals were wide. Total health service use could only be calculated for one-quarter of patients (intervention, *n* = 17; control, *n* = 17) who had complete cost data at all three time points, making the results difficult to interpret. In this small sample of patients, health service costs were higher for the intervention patients but there was overall cost saving in the intervention arm due to the lower costs associated with delivering the physiotherapy treatment in a group setting.

## 4. Discussion

### 4.1. Summary

A total of 136 patients were recruited to the trial (68/68: Intervention/Control). Change in Total SPADI score at 26 weeks from baseline was the primary outcome measure. The initial analysis suggested that there was not equivalence between the groups, favouring the group physiotherapy arm. However, there was a small sample size at week 26, resulting in a wider confidence interval, which may have been the reason for this finding of non-equivalence. On secondary analysis (inputting data from 12 weeks) the groups showed equivalence. The analysis of variance shows that the change in SPADIs score was similar in both groups across the visits. Overall, there is little difference in outcome at 26 weeks and no difference at 52 weeks. It can therefore be concluded that group physiotherapy for SAI is cost saving and does not impact negatively on the health-related quality of life. The main driver of the difference were the physiotherapy costs. Both groups received a similar number of treatments, with the individual care costing £252 per patient treated compared to £84 for six group sessions (based on an average of three patients per class), a cost saving of £168 per patient referred to physiotherapy for shoulder impingement symptoms. The attendance rates were similar at the classes compared with the individual sessions, highlighting that patients will commit to classes as well as to individual treatments.

### 4.2. Strengths and Limitations

We acknowledge several limitations in the study. The recruitment fell short of the power calculation of 200 (100 per group) and this may have contributed to the failure to show equivalence at the primary endpoint. However, any difference in outcome at 26 weeks was toward the group physiotherapy treatment arm and equivalence between treatment groups was shown at 12 weeks.

There were significant withdrawal rates as the study progressed, with few completing the study at 52 weeks. A frequent reason for patients dropping out of the study was to seek alternative management outside the study protocol, for example, repeat corticosteroid injection or ultrasound guided injection. The number of patients receiving further steroid injection was 10 (Intervention = 4, Control = 6) and the number referred for ultrasound investigation with the potential for receiving a guided steroid injection was 32 (Intervention = 19, Control = 13). The high attrition rate meant that only one quarter of patients had complete cost data making the economic analysis results difficult to interpret.

Another consequence of slow recruitment was that we did not achieve our intended size of groups (5–10) in the group physiotherapy, instead achieving three per class. Larger recruitment to classes would have increased the cost saving.

### 4.3. Comparison with Existing Literature

There are several studies comparing individual and group physiotherapy in musculoskeletal conditions [11]. Most of these focus on spinal pain. One study of frozen shoulder [24] showed superior outcomes for a group exercise class over individual physiotherapy. This study suggested that peer support and motivation, as well as fostering behavioural change via a self-management approach, may have explained the difference in outcome. Another study compared physiotherapist supervised group exercises with individual home exercise after a single information session with a physiotherapist in patients with non-traumatic inoperable painful shoulder [25] This study found less functional limitation at five weeks in those performing exercises in a group setting. This current study examines subacromial shoulder pain and no other studies comparing group and individual physiotherapy in this condition were found. Our study has found that for SAI patients group physiotherapy is at least equivalent to individual physiotherapy, with potential for significant economic savings.

Various explanations for the lack of superiority of individual physiotherapy have been offered [11]. Group interventions spend time on exercise and education whereas individual approaches may include more passive approaches. Exercise has been shown to have a positive effect on biopsychosocial factors associated with pain conditions [26] and this may be an effective aspect of both approaches. A group approach may also increase social interaction between patients. This peer support may be an advantage of the group approaches by addressing more of the psychosocial issues associated with painful conditions. A group approach may address the multidimensional biopsychosocial needs of these patients more effectively than an individual approach which may be more focused on physical factors.

### 4.4. Implications for Research and/or Practice

The cost saving of group-based approaches with similar outcomes has implications for commissioning and planning of shoulder pain treatment. With long NHS waiting times for musculoskeletal interventions including for physiotherapy, a group-based approach may increase the capacity of services without compromising clinical effectiveness.

Further research should address which aspects of physiotherapy treatment contribute to its effectiveness in treating shoulder pain. It is not known what should be included in a group-based programme and how much emphasis should there be on education and approaches addressing the biopsychosocial model. For example, can a stratification of care approach identify which patients may benefit more from an individual or group approach to treatment?

Group based physiotherapy treatment for subacromial shoulder pain is therefore a cost saving approach with similar clinical outcome to individual physiotherapy.

### 4.5. How This Fits in to Clinical Practice

Shoulder pain is a common problem in primary care with physiotherapy and subacromial corticosteroid injection commonly used to treat subacromial impingement (SAI). Group based approaches to physiotherapy have been explored in other musculoskeletal conditions but not in SAI when preceded by corticosteroid injection. With increasing engagement of first contact physiotherapists in the primary care multidisciplinary team, exploration of the most cost-effective approach to physiotherapy for SAI is important. In this study we show that group-based physiotherapy for SAI is cost saving compared with individual physiotherapy with similar clinical outcomes.

## 5. Conclusions

There is little difference in outcome between group and individual physiotherapy following corticosteroid injection for SAI at 26 weeks and no difference at 52 weeks. Group physiotherapy was cheaper to deliver per patient. Group physiotherapy for SAI produces similar clinical outcomes to individual physiotherapy with potential cost savings. Due to low recruitment to and high drop out from our study, firm conclusions are difficult and further research is required to give a definitive answer to this research question.

## Figures and Tables

**Figure 1 ijerph-17-05565-f001:**
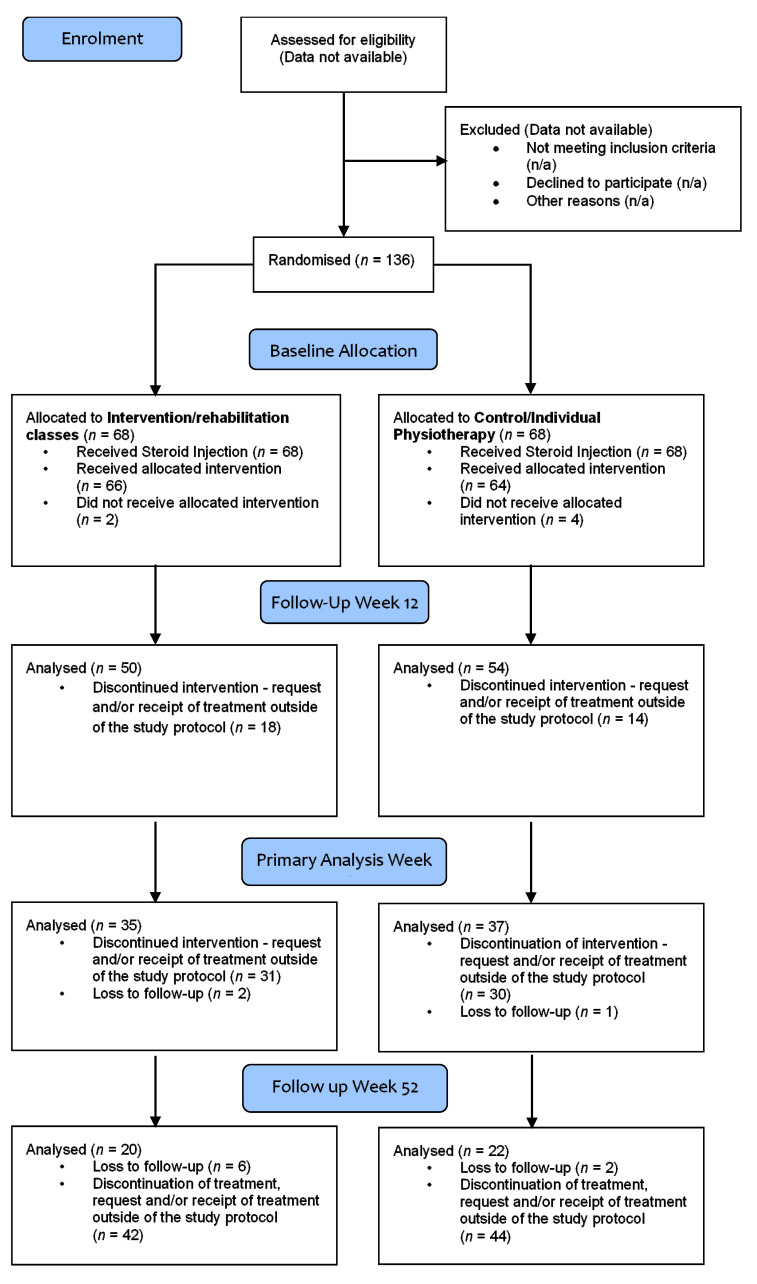
Consolidated Standards of Reporting Trials (CONSORT) diagram.

**Figure 2 ijerph-17-05565-f002:**
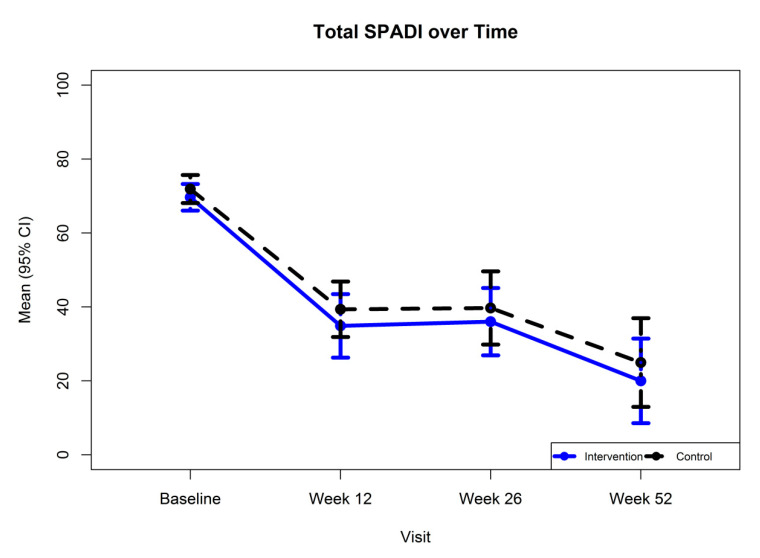
Mean (95% CI) for SPADI Total Score over time. (Intervention—Baseline *n* = 68, Week 12 *n* = 50, Week 26 *n* = 35, Week 52 *n* = 20; Control—Baseline *n* = 68, Week 12 *n* = 54, Week 26 *n* = 37, Week 52 *n* = 22).

**Table 1 ijerph-17-05565-t001:** Inclusion and exclusion criteria.

Inclusion Criteria	Exclusion Criteria
Adult subjects (≥18 years) with unilateral shoulder pain of more than four weeks duration SPADI score of ≥30 Shoulder pain was defined as pain in the shoulder region, including the upper arm, elicited by active or passive shoulder movement. The diagnosis of “subacromial pain” is defined by range as: no limitation in passive range of movement or restriction of passive range of movement mainly in abduction rather than external rotation [13].	Inability to give informed consent Physiotherapy or injection treatment for current shoulder pain in previous three months Blood coagulation disorders Bilateral shoulder pain Evidence of systemic infection Abnormal shoulder X-ray defined as significant glenohumeral or subacromial joint space narrowing suggesting Osteoarthritis of glenohumeral joint or complete Rotator Cuff rupture, Evidence of rotator cuff tear, tested by external rotation lag sign, drop sign, internal rotation lag sign and static muscle resistance in external rotation, internal rotation and abduction History of significant trauma to the shoulder Inflammatory joint disease History of cerebrovascular accident Allergy or contraindication to Triamcinolone/contraindication to injection. Evidence of referred pain from cervical spine disease Pregnancy or breast feeding Patients whose first language is not English

**Table 2 ijerph-17-05565-t002:** Baseline Characteristics.

Variable	Intervention	Control
Number of patients (*n*)	68	68
Mean Age (SD)	54.5 (10.9)	58.1 (11.1)
Gender (*n*, %)		
Female	44 (64.7)	39 (57.4)
Male	24 (35.3)	29 (42.6)
Ethnic origin (*n*, %)		
Caucasian	68 (100)	66 (97)
Black	0	2 (2.9)
Paid employment (*n*, %)		
No	34 (50)	41 (60.3)
Yes	34 (50)	27 (39.7)
Off work due to shoulder pain (*n*, %)	12 (35.3)	6 (22.2)
Lost working days (median, IQR)	14.5 (4–50)	42.5 (24–80)
Right hand dominance (*n*, %)	57 (83.8)	56 (82.4)
Painful shoulder (*n*, %)		
Right	48 (70.6)	42 (61.8)
Left	20 (29.4)	26 (38.2)
Duration of episode (weeks, median, IQR)	30 (21–55)	33.5 (20–80)
Onset of shoulder pain (*n*, %)		
Injury	20 (29.4)	16 (23.5)
Previous episodes of shoulder pain (*n*, %)		
Yes	17 (25)	18 (26.5)
Number of episodes	3 (1–5)	2.5 (1–5)
Time since last episode	12 (7–48)	24 (7–60)
Previous shoulder injection		
Yes (*n*, %)	16 (23.5)	22 (32.3)
No of injections (median, range)	1 (1–3)	1 (1–3)
Improved with injection (*n*, %)	13 (81.2)	13 (59.1)
Duration of improvement (weeks, mean, range)	8 (3–24)	19 (8–50)
Time since last injection (months, mean, range)	10 (7–15.5)	15.5 (7–36)
Previous physiotherapy for shoulder		
Yes (*n*, %)	21 (30.9)	21 (30.9)
Improved with physiotherapy (*n*, %)	13 (61.9)	7 (33.3)
Time since physiotherapy (months, mean, range)	9 (5–48)	15 (9–24)

**Table 3 ijerph-17-05565-t003:** **Shoulder pain and disability index** (SPADI) results–pain, disability and total score.

SPADI Score	Intervention		Control	
	*n*	Mean (SD)	*n*	Mean (SD)
**SPADI–Pain**				
Baseline	68	74.6 (14.2)	68	76.4 (14.7)
Week 12	50	42.6 (32.4)	54	44.7 (29.9)
Week 26	35	43.4 (28.9)	37	44.4 (31.2)
Week 52	20	25.4 (28.2)	22	29.8 (29.7)
**SPADI–Disability**				
Baseline	68	66.6 (16.4)	68	69.1 (17.2)
Week 12	50	30.0 (29.5)	54	36.0 (27.5)
Week 26	35	31.4 (26.3)	37	36.8 (29.8)
Week 52	20	16.6 (23.7)	22	21.9 (26.7)
**SPADI–Total Score**				
Baseline	68	69.7 (14.9)	68	71.9 (15.7)
Week 12	50	34.9 (30.2)	54	39.4 (27.5)
Week 26	35	36.0 (26.5)	37	39.7 (29.7)
Week 52	20	20.0 (24.5)	22	25.0 (27.0)

**Table 4 ijerph-17-05565-t004:** SPADI–Change from Baseline to 26 weeks.

SPADI	Mean (Standard Error)		Mean Difference (SE)
	Intervention	Control	
Pain	−27.6 (4.7)	−30.3 (4.3)	2.7 (6.3)
Disability	−33.6 (4.4)	−31.2 (3.9)	−2.4 (5.8)
Total Score	−31.3 (4.3)	−30.8 (3.8)	−0.43 (5.7)

**Table 5 ijerph-17-05565-t005:** Unit costs of health services (£ UK).

Resource Item	Unit Cost	Details	Source
Practice GP visit	35	Per patient contact lasting 11.7 min. Excluding direct care staff costs & qualifications	Unit costs of Health and Social Care 2014 [21]
Accident & Emergency visit	142	Weighted average of type 01 emergency medicine, admitted and non-admitted.	NHS reference costs 2013–2014 [22]
Physiotherapy attendance	42	Non-Admitted, face to face follow-up attendance.	NHS reference costs 2013–2014 [22]
Rheumatology	92	Non-Admitted, face to face follow-up attendance.	NHS reference costs 2013–2014 [22]
Trauma & Orthopaedic outpatient attendance	105	Consultant led, non-admitted, follow-up face to face attendance	NHS reference costs 2013–2014 [22]
Outpatient Clinic attendance [unspecified]	109	weighted average of all outpatient attendances	NHS reference costs 2013–2014 [22]
Analgesics	various	-	British National Formulary 68 (January 2015) [23]

**Table 6 ijerph-17-05565-t006:** Cost (£) of health service use over the study period by group. Values are means (standard deviations) unless otherwise stated.

Service Use	12 Weeks				26 Weeks				52 Weeks			
	Intervention		Control		Intervention		Control		Intervention		Control	
	*n*	Mean (sd)	*n*	Mean (sd)	*n*	Mean (sd)	*n*	Mean (sd)	*n*	Mean (sd)	*n*	Mean (sd)
Practice GP visit	50	4.2 (18.22)	54	5.19 (19.71)	35	18.00 (73.21)	37	3.78 (11.02)	20	7.00 (31.30)	22	0 (0)
Accident & Emergency visit	50	0 (0)	54	0 (0)	35	8.11 (48.00)	37	0 (0)	20	0 (0)	22	0 (0)
Physiotherapy attendance (non-study treatment)	50	4.2 (29.70)	54	0.78 (5.72)	35	0 (0)	37	0 (0)	20	6.30 (28.17)	22	0 (0)
Outpatient attendances	50	4.2 (20.78)	54	0 (0)	35	21.8 (98.21)	37	0 (0)	20	9.2 (41.14)	22	0 (0)
Mean health services costs	50	12.60 (47.83)	54	5.96 (20.32)	35	47.91 (138.57)	37	3.78 (11.02)	20	31.70 (115.59)	22	0 (0)
Mean difference (bootstrap 95% CI) *	6.64 (−5.48, 21.25)				44.13 (5.38, 96.59)				31.70 (0, 85.75)			

* Percentile confidence intervals based on 1000 bootstrap re-samples. Significance (*p* < 0.05) was judged where the confidence intervals of differential means excluded zero.

**Table 7 ijerph-17-05565-t007:** Service and treatment physiotherapy costs over the 52 weeks study period.

Costs	Intervention		Control		Mean Difference (Bootstrap 95% CI) *
	*n*	Mean (sd)	*n*	Mean (sd)	
Total health service cost	17	79.65 (206.84)	17	2.06 (8.49)	77.59 (−1.62, 178.09)
Physiotherapy treatment cost	68	74.53 (33.64)	68	188.59 (96.01)	−114.06 (−136.73, −89.38)
Overall cost	17	173.53 (206.81)	17	214.53 (83.45) 254.06 (8.49)	−41.00 (−129.20, 79.13)

* Percentile confidence intervals based on 1000 bootstrap resamples.

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
