# Peer review of "The Effectiveness of Individual or Group Physiotherapy in the Management of Sub-Acromial Impingement: A Randomised Controlled Trial and Health Economic Analysis"

_ijerph, 2020, doi:10.3390/ijerph17155565_

Round 1

Reviewer 1 Report

Thank you for the opportunity to review this manuscript. The authors' efforts over many years are recognised. Given this effort, the recruitment and follow up rates are disappointing, particularly given that they substantially undermine the quality of the study (particularly the low follow up rates).

For me, it’s hard to be convinced that the study offers clinicians much useful information. The difference in physio costs appears to be the most robust result, but this maths could have been done without running the study. The clinical outcomes are constrained by small numbers and low follow up rates, but overall, for those assessed it appears that participants realised similar improvements. Whether this improvement can be attributed to the physio program and/or CSI is unknown.

I’ve made a few, non-exhaustive comments below. Overall, there is a sense of author fatigue with the manuscript, particularly with respect to clarify of expression and critical thinking.

INTRODUCTION

“All patients initially received a subacromial corticosteroid injection, which has been shown to give short term pain relief (9), to allow patients to undertake their rehabilitation programme, with individuals randomised to either group based or individual physiotherapy.” This information is relevant for the Method – suggest move there.

METHOD

I think the authors need to justify how SAI is diagnosed, rather than simply stating “The diagnosis of “subacromial pain” is defined by range as: no limitation in passive range of movement or restriction of passive range of movement mainly in abduction rather than external rotation.” At the very least a reference for this definition should be provided and some discussion of reproducibility.

Intervention needs to be described in sufficient detail to allow the reader to assess and reproduce content. Also, I think the intervention is best considered as CSI and physiotherapy, and that the effects of either treatment cannot be differentiated.

“Routine individual physiotherapy: 6 sessions weekly (30 min) for 6 weeks.” Is this correct? Almost daily physio for 6 weeks?

Health care costs must be described in sufficient detail to know what costs were included and who/how this information was collected.

Flow chart: for follow up, did pts discontinue the intervention or not complete follow up?

Intention to treat analysis?

RESULTS

It’s difficult to understand the results of the economic analysis without adequately described methods. For example, GP visits, were they visits for shoulder pain? The costs for physio services are easier to understand, given the explanations provided.

Discussion

“Overall there is little difference in outcome at 26 weeks and no difference at 52 weeks. It can therefore be concluded that group physiotherapy for SAI is cost saving and does not impact negatively on the health related quality of life.” This statement is believable, but seems inconsistent with the abstract which rejected the null hypothesis.

A primary result is discontinuation of treatment and loss to follow up.

Spelling mistakes throughout need to be addressed.

Author Response

Thank you for your review of our submitted paper. I have detailed our responses below and uploaded a revised version of the paper with changed highlighted in red text for your attention. 

INTRODUCTION

“All patients initially received a subacromial corticosteroid injection, which has been shown to give short term pain relief (9), to allow patients to undertake their rehabilitation programme, with individuals randomised to either group based or individual physiotherapy.” This information is relevant for the Method – suggest move there.

We have moved this as suggested.

METHOD

I think the authors need to justify how SAI is diagnosed, rather than simply stating “The diagnosis of “subacromial pain” is defined by range as: no limitation in passive range of movement or restriction of passive range of movement mainly in abduction rather than external rotation.” At the very least a reference for this definition should be provided and some discussion of reproducibility.

We have provided a reference as suggested. 

Intervention needs to be described in sufficient detail to allow the reader to assess and reproduce content. Also, I think the intervention is best considered as CSI and physiotherapy, and that the effects of either treatment cannot be differentiated.

We agree that these cannot be differentiates and should be considered as CSI and physiotherapy and have adjusted the intervention titles to reflect this.

“Routine individual physiotherapy: 6 sessions weekly (30 min) for 6 weeks.” Is this correct? Almost daily physio for 6 weeks?

The text has been adjusted to clarify that physiotherapy was once weekly for 6 weeks.

Health care costs must be described in sufficient detail to know what costs were included and who/how this information was collected.

We have now clarified the text and included a table of unit costs.

Flow chart: for follow up, did pts discontinue the intervention or not complete follow up?

The flow chart has been updated.

Intention to treat analysis?

Analysis was per protocol analysis.

RESULTS

It’s difficult to understand the results of the economic analysis without adequately described methods. For example, GP visits, were they visits for shoulder pain? The costs for physio services are easier to understand, given the explanations provided.

We have now clarified this in the text and included a table of unit costs.

Discussion

“Overall there is little difference in outcome at 26 weeks and no difference at 52 weeks. It can therefore be concluded that group physiotherapy for SAI is cost saving and does not impact negatively on the health related quality of life.” This statement is believable, but seems inconsistent with the abstract which rejected the null hypothesis.

A primary result is discontinuation of treatment and loss to follow up.

The text has been updated to reflect these comments.

Spelling mistakes throughout need to be addressed.

Reviewer 2 Report

In this manuscript, Ryans et al. report results from an RCT + economic analysis on individual or group physiotherapy for subacromial impingement syndrome. The results just missed statistical significance, however there was a tendency for both interventions to be equivalent (or non-inferior), with group sessions being cheaper.

This manuscript is of interest for primary care.

A major ethical limitation was that this RCT was only registrered retrospectively. However, now that it has been registered on clinicaltrials.gov, the results should be reported and published, and I commend the authors for pursuing this. Also non-included patients were not characterized, which is a pitty, but this cannot be done retrospectively.

MAJOR COMMENTS:

1/ Figure 1 needs to be redrawn. There are "paragraph" signs/line breaks visible which need to be removed. Also, the quality of the image needs to be improved to at least 300 DPI (is this a screenshot?)

2/ Please specify who were the two researchers who assessed outcomes (under heading 2.5).

3/ Under heading 2.7., randomisation is not clearly described. What was the size of the blocks? How was randomisation generated, using a computer algorithm? Or you mixed envelopes in a jar and randomly drew them out and numbered them? Please be more specific.

Also there is a typo here: "randomsiations"

4/ Under heading 2.8 sample size calculation: what was the actual percentage of drop-outs considered in this calculation? Also, what was the standard error that you had expected for this calculation?

5/ In section 2.9, you again specify a margin of +/- SPADI points, but this time with a different reference (ref. 23, one paragraph higher you used ref. 19); is this needed?

6/ In Table 2, since you have <100 individuals in each group, please report no more than two significant digits, and add standard error (e.g. for age: 54 +/- SE years vs. 58 +/- years) and add a column were you assess significance (report P-values using t-test/Mann-Whitney or chi-square tests [or other as appropriate, depending on normality of your data distribution], and add these to the statistical methods section).

7/ What was the effect on the primary outcome at 12 weeks? This is not clearly reported.

8/ For the cost-effectiveness, I don't understand why the researchers "assumed that all patients attended six sessions of physiotherapy". Why not use the actual number of sessions each participant attended? And divide the cost of group sessions by actual number of participants in each session? Otherwise the estimates are not accurate.

MINOR COMMENTS:

9/ Please add the NCT clinical trial registration number in the Abstract.

10/ Please specify the method of consenting. You write that the physiotherapist "obtained consent". Was this WRITTEN informed consent? Was this a consent form approved by the Ethical Committee?

Author Response

Thank you for your review of our paper. We have responded to your comments below and uploaded are revised version of our paper with changes highlighted in red text.

MAJOR COMMENTS:

1/ Figure 1 needs to be redrawn. There are "paragraph" signs/line breaks visible which need to be removed. Also, the quality of the image needs to be improved to at least 300 DPI (is this a screenshot?)

This has be amended in the revised paper.

2/ Please specify who were the two researchers who assessed outcomes (under heading 2.5).

In the original protocol 2 observers were available in case of unavailability - the second observer was not required so all observations were undertaken by the same researcher. The text has been updated to reflect this. 

3/ Under heading 2.7., randomisation is not clearly described. What was the size of the blocks? How was randomisation generated, using a computer algorithm? Or you mixed envelopes in a jar and randomly drew them out and numbered them? Please be more specific.

Also there is a typo here: "randomsiations"

This has been addressed and updated in the revised paper. 

4/ Under heading 2.8 sample size calculation: what was the actual percentage of drop-outs considered in this calculation? Also, what was the standard error that you had expected for this calculation?

This has been updated in the revised paper.

5/ In section 2.9, you again specify a margin of +/- SPADI points, but this time with a different reference (ref. 23, one paragraph higher you used ref. 19); is this needed?

References have been updated. 

6/ In Table 2, since you have <100 individuals in each group, please report no more than two significant digits, and add standard error (e.g. for age: 54 +/- SE years vs. 58 +/- years) and add a column were you assess significance (report P-values using t-test/Mann-Whitney or chi-square tests [or other as appropriate, depending on normality of your data distribution], and add these to the statistical methods section).7/ What was the effect on the primary outcome at 12 weeks? This is not clearly reported.

Although we appreciate that randomisation removes selection bias and does not cannot guarantee that groups are equivalent at baseline, any differences that are observed are the result of chance.  According to the CONSORT statement (2010) Section 15 Baseline Data,  significance tests of baseline characteristics are “superfluous and can mislead investigators and their readers”. They recommend that comparisons at baseline should be” based on consideration of the prognostic strength of the variables measured and the size of any chance imbalances that have occurred”..  Furthermore, it is also in keeping with the CONSORT statement to report means and standard deviations for baseline data and not standard errors or confidence intervals as these inferential rather than descriptive statistics. Therefore, in order to adhere to the CONSORT statement, we politely decline to the request to perform hypothesis testing on the baseline characteristics and report standard errors.

8/ For the cost-effectiveness, I don't understand why the researchers "assumed that all patients attended six sessions of physiotherapy". Why not use the actual number of sessions each participant attended? And divide the cost of group sessions by actual number of participants in each session? Otherwise the estimates are not accurate.

This has now been changed to the actual numbers, and the text updated, thanks.

MINOR COMMENTS:

9/ Please add the NCT clinical trial registration number in the Abstract.

Added to absract

10/ Please specify the method of consenting. You write that the physiotherapist "obtained consent". Was this WRITTEN informed consent? Was this a consent form approved by the Ethical Committee?

Text has been updated to reflect written informed consent approved by the ethical committee.